

# Environmental DNA metabarcoding for monitoring metazoan biodiversity in Antarctic nearshore ecosystems

Laurence J. Clarke[1,2], Leonie Suter[1], Bruce E. Deagle[3], Andrea M. Polanowski[1], Aleks Terauds[1], Glenn J. Johnstone[1] and Jonathan S. Stark[1]

[1] Australian Antarctic Division, Kingston, Tasmania, Australia
[2] Institute for Marine and Antarctic Studies, University of Tasmania, Hobart, Tasmania, Australia
[3] Commonwealth Scientific and Industrial Research Organisation, Hobart, Tasmania, Australia

Corresponding author
Laurence J. Clarke,
laurence.clarke@aad.gov.au

## ABSTRACT

Antarctic benthic ecosystems support high biodiversity but their characterization is limited to a few well-studied areas, due to the extreme environment and remoteness making access and sampling difficult. Our aim was to compare water and sediment as sources of environmental DNA (eDNA) to better characterise Antarctic benthic communities and further develop practical approaches for DNA-based biodiversity assessment in remote environments. We used a cytochrome *c* oxidase subunit I (COI) metabarcoding approach to characterise metazoan communities in 26 nearshore sites across 12 locations in the Vestfold Hills (East Antarctica) based on DNA extracted from either sediment cores or filtered seawater. We detected a total of 99 metazoan species from 12 phyla across 26 sites, with similar numbers of species detected in sediment and water eDNA samples. However, significantly different communities were detected in the two sample types at sites where both were collected (*i.e.*, where paired samples were available). For example, nematodes and echinoderms were more likely to be detected exclusively in sediment and water eDNA samples, respectively. eDNA from water and sediment core samples are complementary sample types, with epifauna more likely to be detected in water column samples and infauna in sediment. More reference DNA sequences are needed for infauna/meiofauna to increase the proportion of sequences and number of taxa that can be identified. Developing a better understanding of the temporal and spatial dynamics of eDNA at low temperatures would also aid interpretation of eDNA signals from polar environments. Our results provide a preliminary scan of benthic metazoan communities in the Vestfold Hills, with additional markers required to provide a comprehensive biodiversity survey. However, our study demonstrates the choice of sample type for eDNA studies of benthic ecosystems (sediment, water or both) needs to be carefully considered in light of the research or monitoring question of interest.

## INTRODUCTION

Antarctic benthic communities have high biodiversity and high levels of endemism (*Aronson et al., 2007*), but remain poorly characterised (*Halanych & Mahon, 2018*). For example, Antarctica supports over 10% of the world's described pycnogonida (sea spider) diversity, much of which is endemic (*Barnes & Peck, 2008*). Antarctic benthic biodiversity may have been underestimated to date, as a series of phylogenetic studies highlight the number of cryptic species within lineages previously thought to have circumpolar distributions (reviewed by *Kaiser et al., 2013*).

Antarctic environments are increasingly under pressure from both global and local human activities. Increasing human activity in Antarctica, linked to both national scientific programs and tourism, is likely to impact nearshore environments and benthic communities (*Tin et al., 2008*). 76% of buildings in Antarctica are in the accessible ice-free areas within 5 km of the coast (*Brooks et al., 2019*), increasing the likelihood of human impacts on Antarctic shallow water nearshore environments (*Stark, Kim & Oliver, 2014a*), which are thought to be relatively rare in Antarctica (*Clark et al., 2015*). Antarctic ecosystems, both terrestrial and marine, are particularly susceptible to introduction of non-indigenous and invasive species in the face of ongoing climate change (*Aronson et al., 2007*; *Chown et al., 2012*). Preserving Antarctic benthic communities in the face of increased human activity, invasive species and climate change requires effective tools for rapid, high-throughput biodiversity assessments and monitoring (*Halanych & Mahon, 2018*).

DNA metabarcoding shows promise as a method for rapid biodiversity assessment. Metabarcoding uses high-throughput sequencing of taxonomically-informative DNA barcoding genes to identify the taxa present in a mixed sample (*Hebert et al., 2003*; *Ji et al., 2013*). Metabarcoding can avoid some of the challenges of traditional survey methods, including time-consuming morphology-based identification (*Deagle et al., 2018*; *Leduc et al., 2019*), expensive underwater visual surveys (*Smith et al., 2015*; *Stat et al., 2019*), and less reliance on taxonomic expertise (*Ji et al., 2013*). Metabarcoding is often applied to DNA extracted from environmental samples such as water, soil or sediment, known as environmental DNA or eDNA (*Thomsen et al., 2012*). Environmental DNA in benthic environments can come from excreted faeces, mucous, and gametes; shed cells and scales; degrading tissue (*e.g.*, moults and carcasses) and whole live organisms (*Taberlet et al., 2018*).

Several studies have now applied DNA metabarcoding to study Antarctic benthic communities, typically focussing on the meiofauna (taxa in the size range ∼45–500 μm). Each of these used sediment samples, including deep-sea sediment (*Sinniger et al., 2016*), from either the Antarctic Peninsula (*Fonseca et al., 2017*; *Sinniger et al., 2016*; *Vause et al., 2019*), or across West Antarctic sites (*Brannock et al., 2018*). These studies found environmental DNA from Antarctic sediments is dominated by typical benthic taxa (*e.g.*, nematodes, arthropods, annelids), and sediment eDNA and morphology-based approaches yielded similar findings when comparing two Antarctic Peninsula sites, but with different taxa driving the results (*Vause et al., 2019*). However, the focus on sediment

to date raises the question: can eDNA extracted from water samples be used to characterise Antarctic benthic communities?

A water eDNA approach has several potential benefits relevant to fieldwork in remote and harsh environments like Antarctica. Water samples for eDNA analysis are simpler and faster to collect than sediment cores, particularly important for difficult-to-access sites where time on-site is limited. Similarly, water eDNA can potentially be used to study both soft- and hard-bottom communities, whereas a sediment-based approach is largely restricted to soft-bottom communities. However, it is not clear whether water eDNA samples are as useful as sediment samples for characterising benthic biodiversity. Indeed, *Holman et al. (2019)* found higher DNA-based diversity in nearshore sediment than water eDNA samples at four urban coastal sites in the United Kingdom, and two recent studies found less than 10% overlap between molecular Operational Taxonomic Units (OTUs) from sediment and benthic water samples (*Antich et al., 2020*; *Brandt et al., 2021*). Similarly, *Koziol et al. (2019)* detected similar numbers of animal taxonomic families in water and sediment eDNA samples, but the number of families detected in the different samples types differed across phyla. For example, more polychaete families (annelids) were detected in sediment samples, but more ascidians (tunicates) were detected in DNA extracted from water.

The aim of this study was to compare water and sediment as eDNA sources for rapid, practical biodiversity assessment for nearshore benthic environments to further inform knowledge of these environments in the Vestfold Hills, East Antarctica. We analysed DNA extracted from sectioned sediment cores and filtered-water samples from sites across the Vestfold Hills region, in the first DNA metabarcoding study of East Antarctic benthic communities. The most suitable sample type for DNA-based biodiversity assessment would yield a high proportion of metazoan reads (for efficiency) and the greatest species richness, whilst reflecting the community composition present at the site (which is not examined here). Our study addressed three main questions: (1) Are there differences between sediment and water eDNA samples in terms of (a) number of reads assigned to metazoans, non-metazoans, and unclassified, (b) the number of metazoan species detected, (c) metazoan community composition? (2) Does eDNA-based species richness decrease with sediment depth, similar to trends observed in morphology-based studies (*Filgueiras et al., 2007*; *Vanhove et al., 2009*)? (3) Do environmental parameters (*e.g.*, depth, distance to open ocean) influence community composition in water eDNA? We also discuss the utility of these methods for benthic monitoring and provide recommendations for sampling and marker choice (based on *in silico* analyses) for future applications of DNA metabarcoding.

## MATERIALS & METHODS

### Sample collection

Field work permits were granted by the Australian Antarctic Division (ATEP 19-20-5097). Water column eDNA samples were collected from 12 nearshore locations around Davis station (26 sites, 1–4 sites per location, Fig. 1). Additionally, replicate samples were collected from six sites, although one pair was collected one day apart, indicating day-to-day

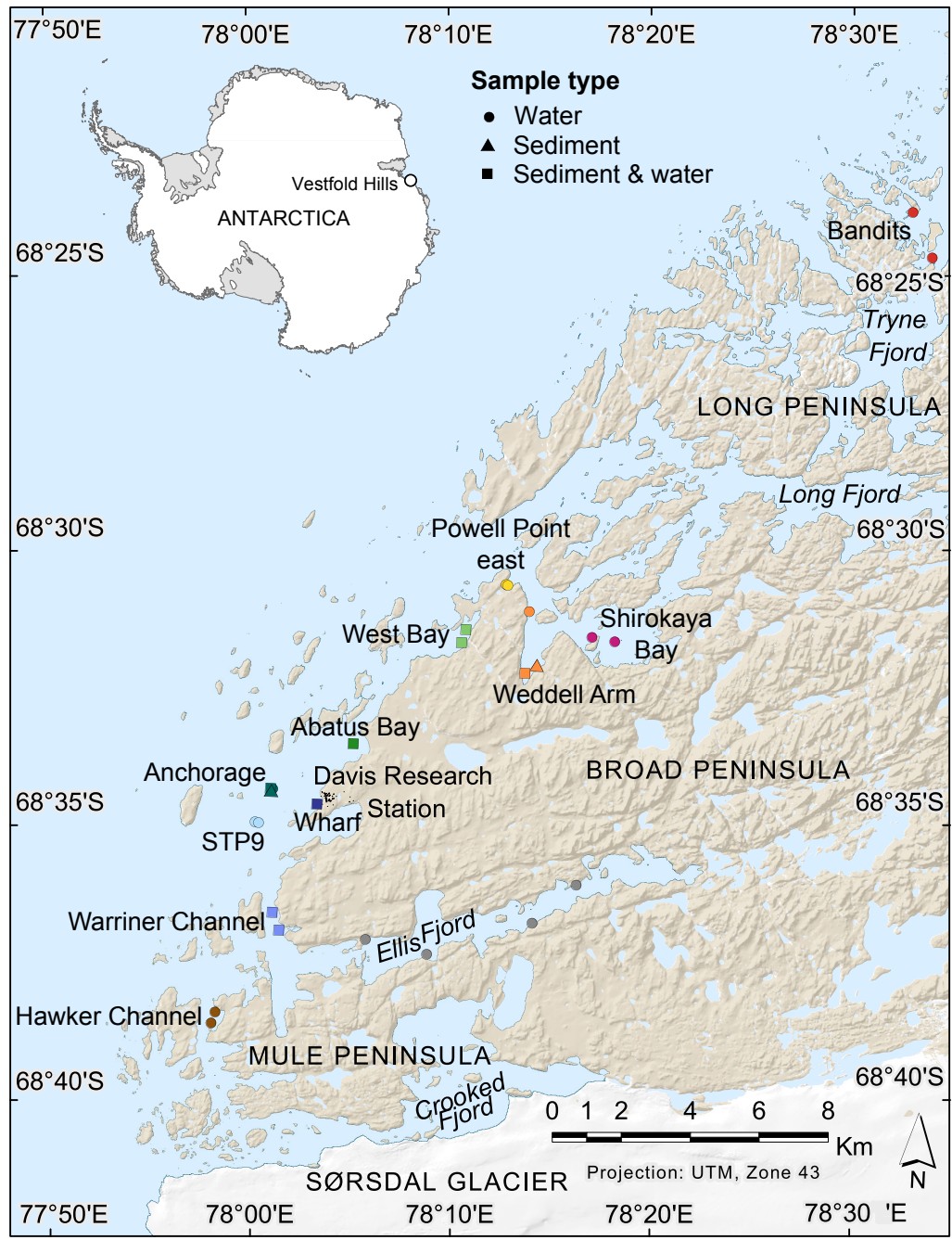

**Figure 1 Map of nearshore benthic sampling sites in the Vestfold Hills, East Antarctica.** Base map produced by the Australian Antarctic Data Centre, and adapted for this study by Helen Achurch (AAD). Sites are coloured as per the ordinations in Figs. 5 and 7.

variability rather than being a true biological replicate. For each sample, 2 L water was collected approximately 1 m from the bottom (depths: 8.4–39 m) using a Niskin bottle. Each water sample was transferred to a 2 L plastic collection bottle and filtered through a Supor 0.45 µM 47 mm membrane filter (Pall Life Sciences, New York, NY, USA) using a Sentino pump (Pall Life Sciences) either in the field or on return to Davis station the same day. Negative controls (2 L Milli-Q water) were also filtered at one site (Bandits) or in the Davis station lab to monitor contamination. After filtration, filter membranes were cut in half and stored at −80 °C as soon as possible. Collection bottles were cleaned with 10% bleach to prevent cross-contamination of samples and rinsed at least twice with Milli-Q water between samples. At the sampling site, bottles were rinsed twice with water from the collection site. Tweezers, filter funnel and the pump base were also cleaned with 10% bleach between samples.

Sediment was collected for genetic analysis from six locations (10 sites, 1–2 sites per location, Fig. 1, Table S1,) using a Kajak gravity corer (KC Denmark A/C, 50 mm core diameter). Water eDNA samples were also collected from all sites with sediment cores, although two of these water samples yielded too few metazoan reads and were not analysed (see Results). Sediment samples from two sites (Abatus Bay and Wharf) were limited to material in the 'core catcher' closing system and were not sectioned. At the eight sites where a true sediment core was obtained, the top of each sediment core was sectioned into 20 mm lengths down to 100 mm (if available) using a fraction tray attached to the top of the core tube, transferred to 250 mL jars and stored at −20 °C for transport to Australia. Sediment samples ($n = 42$) were thawed, homogenized and approximately 250 mg (wet weight) of sediment transferred to Eppendorf tubes for DNA extraction.

## DNA extraction, PCR-amplification and high-throughput sequencing

DNA was extracted from half of each filter and sediment samples using the QIAGEN DNeasy PowerSoil kit. We chose the PowerSoil kit as it provides high quality and quantities of DNA from multiple sample types, including filtered water and sediment (*Hermans, Buckley & Lear, 2018*). We followed the manufacturer's instructions with the exception that samples were homogenized in the PowerBead tube using a FastPrep-24 (MP Biomedicals) at 4.5 m/s for 45 s.

We targeted metazoan (multicellular) organisms by PCR-amplifying part of the mitochondrial cytochrome *c* oxidase subunit I (COI) gene (313 bp for most metazoan species) using the primers from *Leray et al. (2013)*; mlCOIintF: GGWACWGGWT-GAACWGTWTAYCCYCC, jgHCO2198: TAIACYTCIGGRTGICCRAARAAYCA). COI is used as the 'DNA barcode' for metazoans, and benefits from the Barcode of Life Database (BOLD) containing millions of reference COI DNA sequences for over 200,000 animal species (*Hebert et al., 2003*; *Ratnasingham & Hebert, 2007*).

Metabarcoding libraries were prepared for sequencing as per *Clarke et al. (2017)*. Specifically, PCR amplifications were performed in two rounds, the first to amplify the target COI gene and add sample-specific 6 bp multiplex-identifier (MID) tags (forward and reverse primer) and Illumina sequencing primers, the second to add sequencing adapters and additional 8 bp MIDs. Using two rounds of tagging reduces the chance of sequencing
artefacts such as tag-jumping leading to false-positives, as reads must include two pairs of unique MID tags to be assigned to a given sample. The first round was the touchdown protocol as per *Leray et al. (2013)*, namely 95 °C for 10 min, a 16 cycle touchdown phase (62 °C–1 °C per cycle), followed by 25 cycles with an annealing temperature of 46 °C (total of 41 cycles), and a final extension at 72 °C for 5 min. Although using a single low annealing temperature (*e.g.*, 46 °C) rather than the touchdown protocol can increase the number of species detected, this is biased towards increased detection of non-metazoan (non-target) taxa (*Clarke et al., 2017*) and may be particularly problematic for eDNA (*Collins et al., 2019*; *Suter et al., 2020*). We chose the touchdown protocol for this study to maximise the number of metazoan reads per sample. Each reaction mix contained 0.5 µM each of forward and reverse primer, 2 µg BSA, AmpliTaq Gold$^{TM}$ 360 Master Mix in 1 x reaction buffer (Life Technologies, Melbourne, Australia), and 1 µL DNA extract (undiluted) in a total reaction volume of 10 µL. PCR products were diluted 1:10 and Illumina sequencing adapters were added in a second round of PCR (10 cycles with an annealing temperature of 55 °C) using the same conditions as the first round, except primer concentrations were reduced to 0.1 µM each. Products from each round of PCR were separated by electrophoresis and visualized on 2% agarose gels. Equal volumes of second round PCR products were pooled then purified using Agencourt AMPure XP beads (Beckman Coulter, Brea, CA, USA) and the size distribution and concentration of the library assessed on a 2100 Bioanalyzer (Agilent Technologies, Santa Clara, CA, USA). The pool was diluted to 2 nM and paired-end reads generated on a MiSeq (Illumina, San Diego, CA, USA) with the MiSeq Reagent Kit v2 (2 × 250 bp).

## Bioinformatics

Our approach follows that in *Suter et al. (2020)*. In brief, the MiSeq used the 8 bp MIDs to assign sequences to sample-specific FASTQ files (available in the NCBI database under BioProject: PRJNA720767). Paired reads were merged using USEARCH v10.0.240 (*Edgar, 2010*). Only reads with exact matches to first round 6 bp MID tags and COI primer sequences (identified with the R package 'ShortRead', *Morgan et al., 2009*; *R Core Team, 2017*) were kept, as non-exact matches are more likely to contain sequencing errors in the remainder of the sequence. These processed sequences were then pooled and dereplicated using the USEARCH command "fastx_uniques". The USEARCH command "unoise3" was used to remove sequencing errors and chimaeras and create a list of unique zOTU (zero-radius operational taxonomic unit) sequences with a minimum abundance of 8 reads. A zOTU table, with reads for individual samples assigned to unique zOTUs, was created using the USEARCH command "otutab".

We used the 'blastn' command (*Madden, 2013*) to search the zOTU sequences against the NCBI nucleotide database (settings: -num_descriptions 50 -num_alignments 50 -num_threads 8 -perc_identity 80; excluding environmental samples, metagenomes and unidentified organisms using the command -negative_seqidlist). We imported these blastn results into MEGAN to assign taxonomy (*Huson et al., 2016*); LCA parameters "min score": 300; "top percent": 5; "min support": (1). zOTUs that were assigned to contaminant species were removed (only *Homo sapiens* in this dataset). zOTU assignments to metazoan taxa

were manually curated, as LCA parameters (*e.g.*, "min score": 300) can result in zOTUs being assigned to species- or genus-level despite low pairwise sequence identity (*e.g.*, <90%). zOTU sequences with pairwise sequence identity <90% were re-assigned to order level where necessary. The presence of additional metazoan species within zOTUs that were only resolved to a high taxonomic level was investigated by aligning all metazoan zOTU sequences with MUSCLE (*Edgar, 2004*) and creating a consensus neighbour-joining phylogenetic tree using Geneious version 8.1.7 (https://www.geneious.com, Fig. S1). zOTU sequences that were only resolved to a high taxonomic level (*e.g.*, kingdom, phylum, class or order) that formed a closely related (>97% identity) monophyletic group of at least two sequences resembling a species were renamed to *e.g.*, "Metazoa sp. A". These sequences were also searched against all barcode records in the online Barcode of Life Database (BOLD, *Ratnasingham & Hebert, 2007*) Identification System to check if the taxonomy could be resolved to a lower level. Metazoan zOTU sequences that were not closely related to any other sequence, and with <85% identity to a known species, were not included in the species-level analysis.

Three of the 50 most abundant zOTUs not assigned taxonomy using the NCBI database had matches with 80% pairwise identity to metazoan taxa in BOLD (Cyclopoida sp.) that formed a monophyletic group and were assigned to phylum level; three matched non-metazoan taxa; the remaining 44 zOTUs could not be classified. Given the low proportion of zOTUs not assigned using NCBI that were assigned taxonomy with BOLD, and the need for time-consuming manual curation of BOLD search results, we did not search the remaining unclassified zOTU sequences against BOLD.

## Statistical analysis

The metazoan species table was used for all alpha- and beta-diversity analyses. We examined species richness (alpha-diversity) for each sample type with rarefaction curves generated using the 'iNEXT' R package (*Hsieh, Ma & Chao, 2016*), and tested for differences in number of species detected between sediment core sections, or eDNA and the uppermost (surface) core section, using one-way ANOVA and a paired $t$-test, respectively, in R. We tested whether taxa from each phylum were more prevalent in water or sediment samples by fitting a binomial generalised linear model to the presence-absence data with terms for species and sample type (but no interaction term). The species term allows the frequency of occurrence to differ for each species within the taxonomic group, and the sample type term allows an additive effect of sample type that applies equally to all species within the group. We tested the significance of the sample type term using a likelihood ratio test, with a Bonferroni-corrected $p$-value based on the number of test phyla (12).

The most appropriate community dissimilarity measure should yield reproducible results for replicate samples. Therefore, we tested whether replicate water eDNA samples were more similar based on presence-absence data or taking relative sequence abundance into account (binary Jaccard vs. Bray–Curtis dissimilarity) using the Adonis method (*Anderson, 2001*) (compare_categories.py, 999 permutations) in QIIME v1.8.0 (*Caporaso et al., 2010*) (beta_diversity_through_plots.py) based on a rarefied metazoan species table (1,000 reads). The same method was used to explore differences in sediment and water

eDNA community composition using binary Jaccard distance. The analysis was repeated using only water eDNA and surface sediment samples from sites where sufficient data (>1,000 metazoan reads) was available for both sample types (paired sites). The Linear Discriminant Analysis (LDA) Effect Size (LEfSe, (*Segata et al., 2011*)) method was used with default settings (LDA threshold = 2.0, α = 0.05) to highlight taxa that showed different abundances between sediment and water eDNA samples at these paired sites.

Correlations between environmental variables (depth, distance to open ocean) and water eDNA community composition were explored using the *envfit* function in the R package 'vegan' (*Oksanen et al., 2019*). Depth and distance to open ocean were considered gross measures, serving as proxies for finer scale environmental variables such as light level, current speed, propagule sources, freshwater influence, etc., that are most likely to determine faunal distributions. In the absence of data for these finer scale environmental measures, these simple proxies allow us to begin testing for patterns in the data.

### *In silico* PCR

Given we use a single COI marker to generate our dataset, it is unlikely to provide a comprehensive survey of the benthic community. We used *in silico* PCR implemented through the ECOPCR program (*Ficetola et al., 2010*, see Supplemental Information) to inform metabarcoding marker choice for future, potentially multi-marker, eDNA studies of Antarctic benthic communities. Specifically, we explored taxonomic coverage, resolving power and suitability for several metabarcodes targeting mitochondrial COI and 16S genes as well as the nuclear 18S rRNA gene for taxonomic groups under-represented in our COI data (benthic arthropods) and other diverse Antarctic benthic metazoan taxa (annelids and molluscs). Further details are provided in the Supplemental Information.

## RESULTS

### Sediment *vs.* water eDNA
#### *Metazoan reads*
COI metabarcoding of the 80 samples (including 35 water eDNA, 42 sediment and three field or lab negative controls) yielded 688,946 unique sequences prior to denoising, and a total of 5.45 million paired-end reads after filtering and quality control, representing 3950 zero-radius OTUS (zOTUs). 518 zOTUs and 2.163 million reads were assigned to a total of 99 metazoan taxa representing 12 phyla. Approximately 40% of reads were assigned to 'Metazoa' for both sediment and water eDNA samples (Fig. 2). In contrast, almost half (49.5%) of sediment reads were not classified, but only 33% were not classified for water eDNA samples. Lab and field negative controls contained no metazoan zOTUs, except for one with 49 reads assigned to Australian fur seal, likely to represent lab contamination. This species was not detected in any other sample. Sediment samples yielded almost twice the total number of metazoan reads compared to water eDNA samples (Fig. 2), due to both the greater number of reads per sample on average (sediment mean ± SD: 80,800 ± 58,400, water eDNA: 58,700 ± 55,000) and the greater number of sediment samples. Metazoan zOTU sequences that were not closely related to any other sequence, and with <85% identity to a known species, were removed ($n = 39$). The 99 metazoan taxa in the 'species'
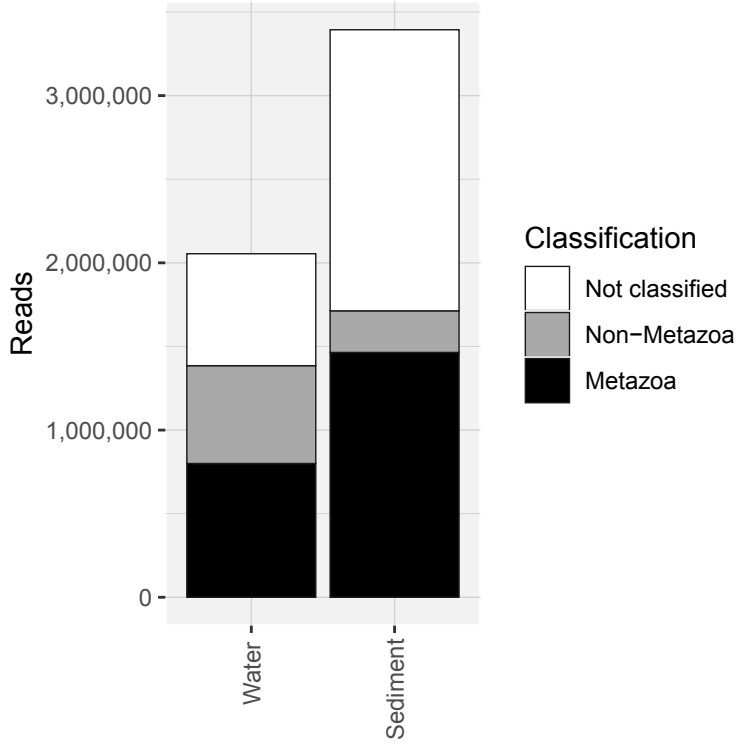

**Figure 2** Number of sequencing reads from the COI marker classified as "metazoan", "non-metazoan" and "not classified" for sediment and water environmental DNA (eDNA) and samples.

table included 479 zOTUs and 2.156 million reads. More than half the metazoan zOTUS (286) and 43% of reads (980,071 reads) were assigned to the copepod *Paralabidocera grandispina*. These zOTUs could also represent *P. antarctica*, a closely related, co-occurring species without a publicly available COI sequence. Eleven samples (five water and six sediment eDNA) had less than 1000 metazoan reads and were excluded from further analysis. The remaining 66 samples showed a large range in the number of reads per sample (1,456–156,765, mean ± SD: 32,671 ± 32,842 reads).

### Species richness

Despite the low number of reads per sample compared to microbial studies, the number of species detected per sample approached an asymptote with 1000 reads for both sediment and water eDNA samples (Fig. 3). For sediment samples, more metazoan taxa were detected in the uppermost core sections, but this was not statistically significant ($F_{4,31} = 1.63$, $P = 0.19$, Fig. 3). For the eight sectioned cores, more than half of all taxa detected in each were typically present in the uppermost core section (mean ±SD % in uppermost section: 67 ± 14%). Combining the uppermost core section with either the second uppermost or bottom core section increased the proportion to approximately 80% of taxa in each core. Similar numbers of taxa were detected in water eDNA samples and the top section of

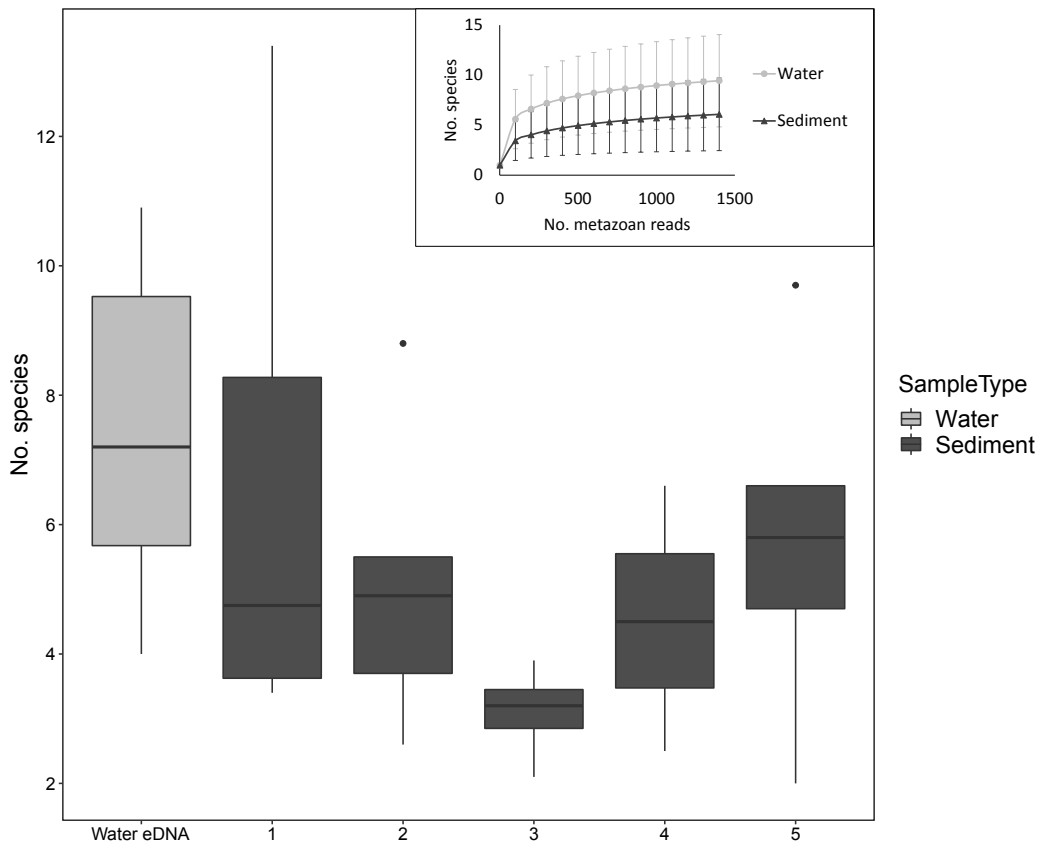

**Figure 3   Number of metazoan species detected per sample by sample type and sediment core section at sites where both sample types were collected, based on 1000 metazoan reads per sample.** The inset shows sample-size based rarefaction curves for all samples. Core samples are classified by core section (1: 0–2 cm, 2: 2–4 cm, 3: 4–6 (or 4–7) cm, 4: 6–8 (7–9) cm, 5: 8–10 (9–11) cm). Cores that were not sectioned are grouped with 0–2 cm sections.

sediment cores at sites where both were collected (paired $t$-test, $t = 0.45$, d.f. $= 7$, $P = 0.67$, Fig. 3).

### Community composition

PERMANOVA showed replicate water eDNA samples (six sites, four locations, $n = 12$) were grouped according to site and location with binary Jaccard distance ($F_{5,6} = 2.91$, $R^2 = 0.71$, $P < 0.001$, Fig. S2), but comparatively weakly for Bray–Curtis dissimilarity based on both rarefied sequence counts ($F_{5,6} = 1.83$, $R^2 = 0.60$, $P = 0.037$) and square-root transformed and rarefied sequence counts ($F_{5,6} = 2.30$, $R^2 = 0.66$, $P = 0.006$). All subsequent beta-diversity analyses were based on binary Jaccard distance as it yielded more reproducible community profiles for replicate samples. For three sites, the majority of metazoan taxa were detected in both replicates (61–68% based on the non-rarefied data), but only 27–38% of metazoans were detected in both replicates at the other three sites. More taxa were always detected in the replicate with greater sequencing depth.

Almost half of all metazoan species were detected in both sediment and water eDNA samples (47/99, 47.5%, Fig. 4). 22 and 30 species were detected exclusively with sediment or water eDNA, respectively. Taxa from four phyla were significantly more prevalent in either water (Echinodermata, Cnidaria and Chordata) or sediment samples (Nematoda, $P < 0.001$ for each phylum). For example, the frequency of occurrence for all echinoderm and chordate taxa, including the Antarctic ploughfish *Gymnodraco acuticeps*, was higher in water than sediment samples, whereas 12/15 nematode taxa had a higher frequency of occurrence in sediment. Metazoan communities from water and sediment eDNA samples were significantly different ($F_{1,64} = 9.59$, $R^2 = 0.130$, $P < 0.001$, Fig. 5A). This analysis includes water samples from hard-bottom sites, where sediment was not sampled, but are likely to support benthic communities that are distinct from soft-bottom sites (*Kirkwood & Burton, 1988*). Similarly, water eDNA samples are likely to be most similar to the uppermost (0–2 cm) sediment samples rather than deeper core sections. However, repeating the analysis with only water eDNA and the uppermost sediment samples from sites where sufficient data (>1,000 metazoan reads) was available for both sample types (paired sites) demonstrated that the two sample types still yielded distinct communities ($F_{1,12} = 2.16$, $R^2 = 0.152$, $P = 0.009$, Fig. 5B). Based on the sediment and water eDNA samples from sites where both were collected, 10 taxa were significantly more abundant in either water eDNA (9) or sediment (1, Fig. 6). Taxa that showed higher relative abundance in water eDNA samples included the Antarctic sea urchin *Sterechinus neumayeri*, an unidentified arthropod, a hesionid polychaete, and the epibenthic polychaete Polynoidae sp. 212 RG-2014 (Fig. 6). The one taxon enriched in sediment samples was the abundant sea-ice associated copepod *Paralabidocera grandispina/antarctica* (Fig. 6).

**Environmental influence on water eDNA community composition**

An nMDS of water eDNA samples showed nearby sites often yielded similar community profiles, as sites within a location were often similar (*e.g.*, two sites near Bandits, and two sites within Shirokaya Bay), as well as sites from nearby locations (*e.g.*, Hawker Channel and Warriner Channel, Fig. 7). However, some locations showed high site-to-site heterogeneity in community composition, *e.g.*, within Ellis Fjord and within Weddell Arm. Different communities are expected between Ellis Fjord sites, as they are spread over several kilometres (Fig. 1) and both environmental conditions and habitat type vary over the length of the fjord (*Kirkwood & Burton, 1988*). Similarly, the two Weddell Arm sites represented distinct habitat types (predominantly hard vs. soft bottom) with different communities. Distance to open ocean explained 17.3% of variance in water eDNA communities but was not significant at the 0.05 level ($R^2 = 0.173$, $P = 0.072$, Fig. 7). There was no evidence that depth was correlated with water eDNA community composition ($R^2 = 0.076$, $P = 0.35$).

*In silico* **PCR**

The Leray COI primers used in our study (*Leray et al., 2013*) provided 88% species-level coverage for benthic arthropods *in silico* (Table 1), with a 4 and 9% increase in coverage using the modified forward primer proposed by *Rennstam Rubbmark et al. (2018)* and *Wangensteen et al. (2018)*, respectively. The increased coverage provided by the modified

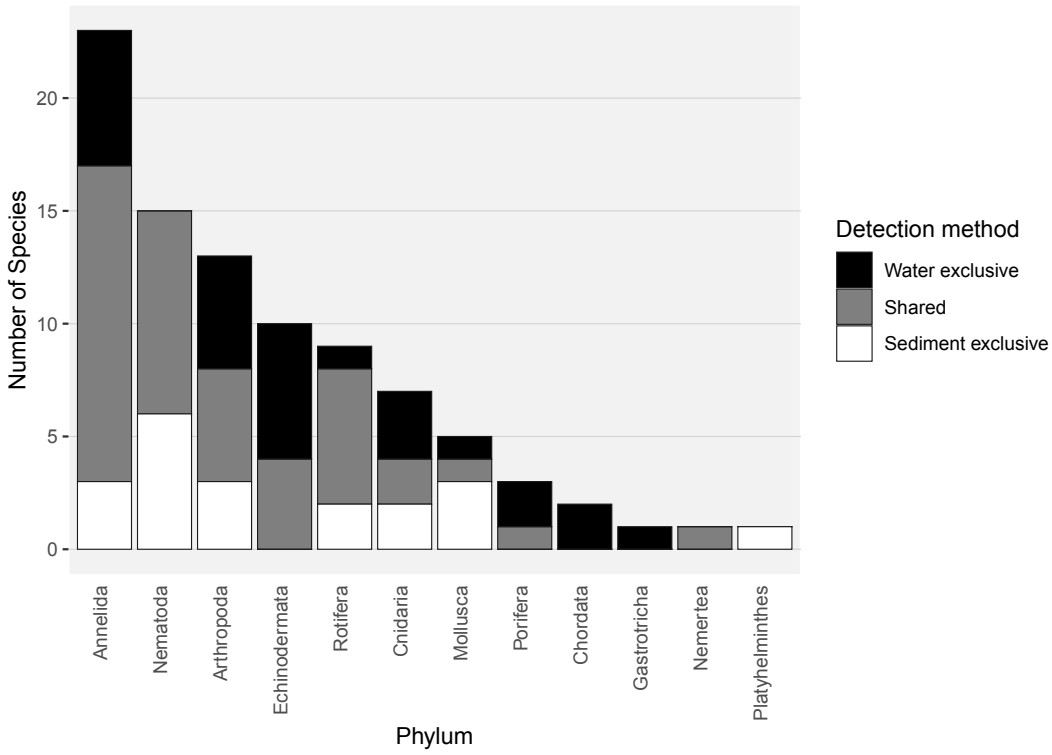

**Figure 4 Number of species per metazoan phylum detected exclusively with water eDNA, sediment, or with both methods.**

COI primers points to the potential to design new COI primers to better target benthic arthropods. All three COI primers resolved 97% of arthropod species 'amplified'.

18S primers targeting the V9 region for Peracarida (including amphipods, isopods, tanaids and cumaceans, *Taberlet et al., 2018*) resolved the lowest proportion of species (79%), and were only able to resolve many taxa at the family or order-level. In contrast, eukaryote 18S primers targeting the V4 region (*Piredda et al., 2017*; *Stoeck et al., 2010*) provided similar taxonomic coverage to other markers (92%), and near-perfect species-level resolution (99%). However, the length and size variation of the amplicon (mean ± SD: 580 ± 88 bp, range: 372–965 bp) make it difficult to sequence with most commonly used sequencing platforms (*e.g.,* Illumina MiSeq), and is likely to introduce a PCR-amplification and sequencing bias against taxa with longer amplicons. 18S rRNA primer-binding sites are also well-conserved across eukaryotic taxa, increasing the likelihood of amplifying non-metazoan eukaryotes.

The two longer 16S metabarcodes resolved a similar proportion of species as the COI metabarcodes (97%, Table 1), with the shorter 16S metabarcode (~57 bp) only resolving 89% of species. Comparatively low species coverage (76%) for the Ins_16S primers (*Clarke et al., 2014*) suggest these primers should be redesigned prior to use with benthic arthropod communities.

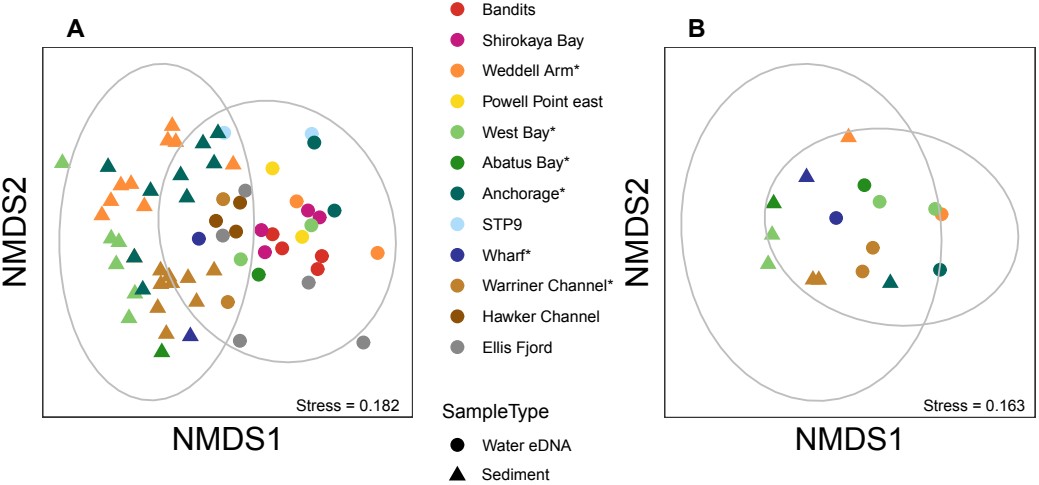

**Figure 5 Sediment and water eDNA samples yield distinct benthic metazoan communities.** Non-metric multidimensional scaling (nMDS) plots based on binary Jaccard distances for metazoan communities from sediment and water eDNA samples based on either (A) all samples, or (B) the uppermost sediment sample and water eDNA for locations where both were collected (marked with an asterisk in the legend). Data ellipses drawn based on a multivariate normal distribution with a confidence level of 0.95.

*In silico* analyses for two other major benthic taxonomic groups (annelids and molluscs) yielded similar results (Tables S3 and S4). Namely, the modified COI primers proposed by *Rennstam Rubbmark et al. (2018)* and *Wangensteen et al. (2018)* increased coverage by 3–4 and 9–11% compared to the Leray primers for annelids and molluscs, respectively. COI primers provided >92% species-level resolution for both groups, whereas resolution was only 22–74% for 18S (both V4 and V9) and 16S rRNA markers. The 18S V9 marker tested (Euka03) also had poor taxonomic coverage for both annelids and molluscs (21 and 25%, respectively). Both 16S markers tested were short (mean length: 63–64 bp), and primers targeting longer variable regions of the mitochondrial 16S gene may have better resolving power.

## DISCUSSION

In this study, we found that water and sediment eDNA samples yielded similar proportions of reads assigned to metazoans and similar metazoan species richness, but differed in their community composition. Our results support the findings of previous eDNA studies that the choice of sample type (*e.g.*, sediment vs. water) needs to be carefully considered in the context of the research or monitoring question (*Antich et al., 2020*; *Brandt et al., 2021*; *Holman et al., 2019*; *Koziol et al., 2019*). For example, is the aim to monitor change in a particular (pelagic or benthic) habitat, or detect specific target taxa, or generate a comprehensive biodiversity assessment for a site? Although more than half the species in this study were detected in both sediment and water eDNA, significantly different metazoan communities were detected in the two sample types at sites where both were collected, highlighting the complementarity of sediment and water eDNA samples and the

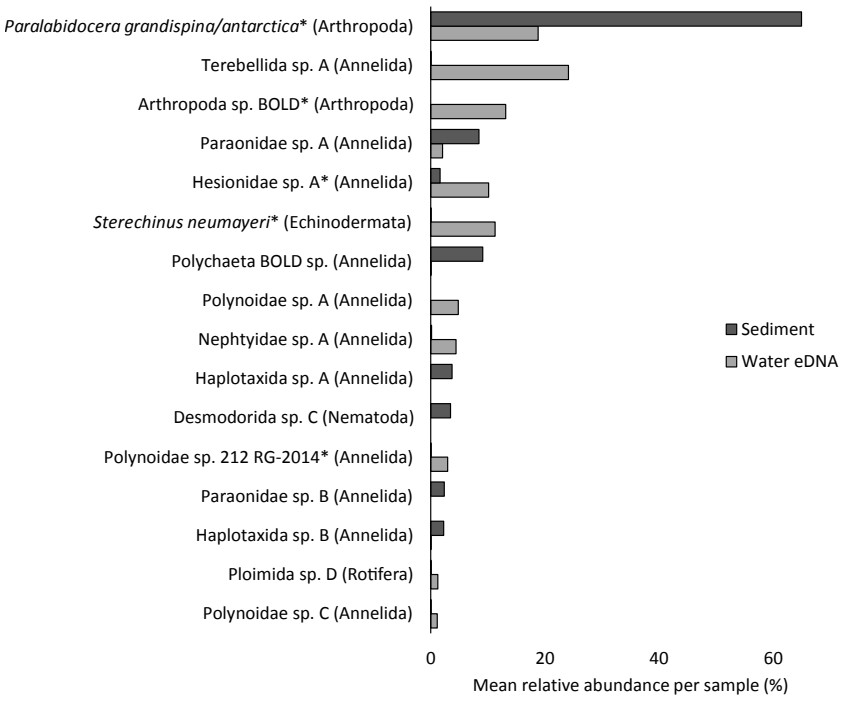

**Figure 6 Mean taxon relative abundance per sample for sediment and water eDNA samples with more than 1,000 reads ($n = 66$).** Taxa with greater than 1% mean relative abundance in either sample type (representing 96% of metazoan reads) are shown. Taxa that were significantly more abundant in the paired water eDNA or sediment samples based on Linear Discriminant Analysis (LDA) Effect Size (LEfSe) are indicated with an asterisk.

utility of collecting both when assessing biodiversity in nearshore benthic environments. The fact that many species were detected in both water and sediment is unsurprising, as some of these species would be found in both sediment and hard-bottom habitats (*e.g.*, polynoids, asteroids, holothurians). In addition, Antarctic benthic ecosystems are often a mosaic of hard and soft habitats, with patch sizes ranging from very small (sub metre) to much larger patches (*Stark, Riddle & Simpson, 2003*). Even where habitat is primarily hard bottom, patches of soft sediments can be found between boulders and on small terraces, and hard surfaces are often overlain by a thin layer of fine sediments due to very low current velocities.

## Influence of sample type on taxa detected

Quantitative comparisons of species richness detected in very different sample types are challenging. However, we found similar numbers of species were detected with both sediment and water eDNA, but the taxa detected in each sample type reflected their respective habitat preferences. Nematode species were more prevalent in sediment, and echinoderms, in particular holothuroids, in water eDNA, likely attributable to habitat preference as infauna and epifauna, respectively, as well as the potential for broadcast spawning of echinoderm gametes. Similar effects of habitat on the taxa detected have been observed in comparisons of sediment and water eDNA from temperate locations (*Holman*

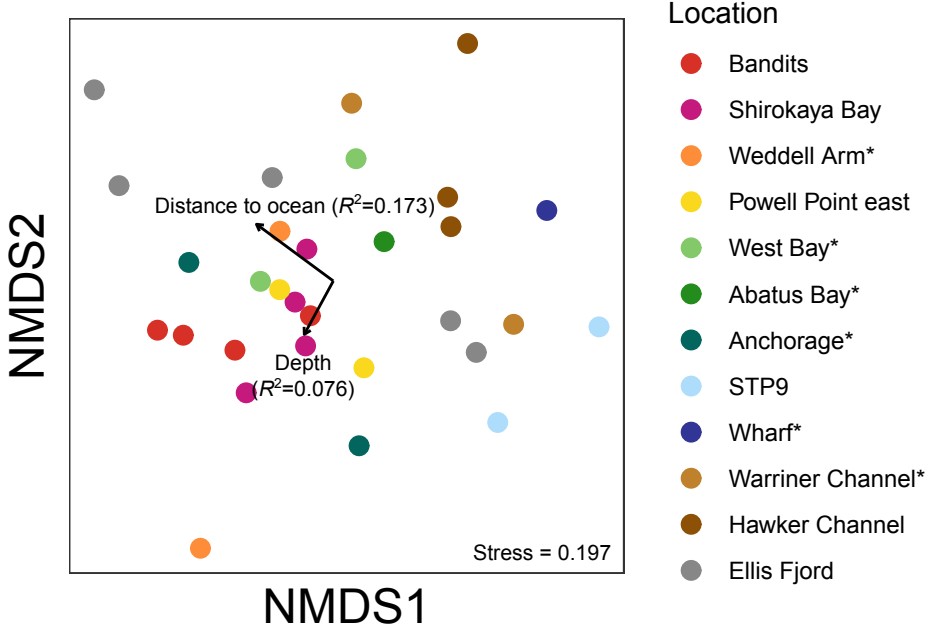

**Figure 7** **Non-metric multidimensional scaling (nMDS) plots based on binary Jaccard distances for metazoan communities from water eDNA samples.** Vectors show correlations with environmental variables, with vector length proportional to the strength of the correlation.

*et al., 2019*; *Koziol et al., 2019*). *Holman et al. (2019)* found more species were detected in sediment than water eDNA samples from coastal sites in the United Kingdom. Both studies filtered similar volumes of water per sample, but *Holman et al. (2019)* extracted DNA from 10 g sediment, compared to 250 mg in this study, which could lead to the greater diversity in sediment samples observed in their study. Methods to concentrate metazoan taxa from larger volumes of either water (*e.g.*, *Suter et al., 2020*) or sediment (*e.g.*, *Brannock & Halanych, 2015*) will also increase the proportion of metazoan reads and number of metazoan species detected, albeit with an increase in the time required to process each sample. Our decision to extract DNA from 250 mg sediment was driven by a practical consideration to use the same extraction kit for water and sediment samples (*Hermans, Buckley & Lear, 2018*). We found pooling data from the uppermost core section with other individual core sections only provided an increase from 67% to approximately 80% of taxa in each core, suggesting we were not missing substantial metazoan diversity by processing 250 mg samples. Similarly, given Antarctic sediment samples are difficult to obtain, they are also often in demand for multiple types of analyses and therefore the use of smaller sample sizes is often preferred. Extracting DNA from small-volume sub-samples may be more realistic in such circumstances. Optimal sediment sampling design, sample size and/or sample processing for eDNA analysis is worthy of further investigation.

Environmental DNA signals are more homogeneous in the water column compared to soil, sediment or settlement plates (*Andersen et al., 2012*; *Koziol et al., 2019*; *Thomsen et al., 2012*), and water eDNA may provide a simple means to assess site-wide biodiversity. For

Clarke et al. (2021), *PeerJ*, DOI 10.7717/peerj.12458

**Table 1  Primers, taxonomic coverage and resolution (species-level) of metabarcodes estimated by *in silico* PCR against a database of benthic arthropod mitochondrial genomes or complete 18S rRNA gene sequences.**

| Name | Locus | Primer sequence (5′–3′) | Mean ± SD length (bp) (min.–max.) | Species-level coverage | Species-level resolution | Reference |
|---|---|---|---|---|---|---|
| Leray COI | COI | GGWACWGGWTGAACWGTWTAYCCYCC TAIACYTCIGGRTGICCRAARAAYCA | 313 ± 1 (307–313) | 69/78 (88%) | 67/69 (97%) | *Leray et al. (2013)* |
| Sauron COI | COI | GGDRCWGGWTGAACWGTWTAYCCNCC TAIACYTCIGGRTGICCRAARAAYCA | 313 ± 1 (307–313) | 72/78 (92%) | 70/72 (97%) | *Rennstam Rubbmark et al. (2018)* |
| Leray-XT | COI | GGWACWRGWTGRACWITITAYCCYCC TAIACYTCIGGRTGICCRAARAAYCA | 313 ± 1 (307–313) | 76/78 (97%) | 74/76 (97%) | *Wangensteen et al. (2018)* |
| V4_18S | 18S, V4 | CCAGCASCYGCGGTAATTCC ACTTTCGTTCTTGATYRATGA | 580 ± 88 (372–965) | 177/192 (92%) | 175/177 (99%) | *Stoeck et al. (2010)*, *Piredda et al. (2017)* |
| Pera02 | 18S, V9 | CCCTTTGTACACACCGCC ATGATCCTTCCGCAGGTTCA | 147 ± 17 (131–231) | 125/192 (65%) | 99/125 (79%) | *Taberlet et al. (2018)* |
| Arth02 | 16S | GATAGAAACCRACCTGGYT AARTTACYTTAGGGATAACAG | 142 ± 1 (139–146) | 75/78 (96%) | 73/75 (97%) | *Taberlet et al. (2018)* |
| Amph01 | 16S | TTRYNACCTCGATGTTGAATT GGTYTGAACTCARATCATGTA | 57 ± 1 (56–61) | 76/78 (97%) | 68/76 (89%) | *Taberlet et al. (2018)* |
| Ins16S_1 | 16S | TRRGACGAGAAGACCCTATA TCTTAATCCAACATCGAGGTC | 194 ± 14 (153–212) | 59/78 (76%) | 57/59 (97%) | *Clarke et al. (2014)* |

example, *Koziol et al. (2019)* found less variation between replicate water eDNA samples compared to sediment or settlement plate replicates. However, recent studies show that marine eDNA signals reflect local communities, with minimal transfer of eDNA between three distinct habitat types separated by less than 5 km subject to tidal and long-shore currents, and less than 10 m in a stratified water column (*Jeunen et al., 2019*; *Jeunen et al., 2020*). East Antarctic soft-bottom communities can show significant small-scale patchiness (*Stark, Riddle & Simpson, 2003*). Such heterogeneity needs to be considered when choosing whether to process one large or many small sediment DNA samples, (*e.g.*, one 10 g sample or five 250 mg samples), or combining multiple small samples into a larger composite sample.

## Reconciling DNA-based and morphology-based approaches

Reconciling a DNA-based approach to biodiversity assessment and monitoring with traditional morphology-based approaches can be difficult (*e.g.*, *Deagle et al., 2018*). Environmental DNA approaches can detect organisms from the full size spectrum, including macro-organisms. For example, we detected Antarctic ploughfish (*Gymnodraco acuticeps*) in one water eDNA sample. This can complicate comparison with traditional morphology-based approaches that would typically focus on a particular size fraction (*e.g.*, meiofauna) or habitat (*e.g.*, epifauna versus infauna). Higher diversity often reported for DNA-based approaches could reflect sampling a broader pool of species. Similarly, environmental DNA cannot differentiate between planktonic (and potentially temporary resident) and benthic organisms (usually a permanent resident of a site). For example, the polychaete Polynoidae sp. 212 RG-2014 was first described from West Antarctic meroplankton communities (*Gallego, Lavery & Sewell, 2014*), and was significantly more abundant in water eDNA compared to sediment samples in this study (Fig. 6). Lastly, a benefit, but also a drawback, of a DNA-based approach is that, although DNA can potentially identify juvenile and larval life stages that may be impossible to identify based on morphology, DNA does not provide demographic information, and cannot differentiate between larval, juvenile and adult life stages. It is possible that DNA-based approaches may detect gametes or eggs of a species at a site where adults are not present. However, such sensitivity could be valuable for the early detection of non-indigenous and invasive species.

Many arthropod taxa present in the Vestfold Hills nearshore benthos were not detected in this study, suggesting that additional markers are necessary for a comprehensive survey of the metazoan community. Taxa not detected, but known to be present in nearshore benthic communities in the Vestfold Hills region (*Everitt, Poore & Pickard, 1980*), included ostracods, gammarid amphipods and isopods. Similarly, only a single zOTU was assigned to an amphipod, typically a common and diverse group in this region, where they can comprise over 80% of the macrofauna (infauna > 0.5 mm, *Stark, Kim & Oliver, 2014b*; *Stark, Riddle & Simpson, 2003*). The failure to detect these taxa could be due to either primer bias or the lack of publicly available reference COI sequences. Available ostracod, isopod and gammarid amphipod mitochondrial genomes contain 1-4 mismatches to the forward Leray primer, and 0-1 mismatches in the reverse, raising the possibility of primer bias.

Crustacean and/or taxon-specific COI or 16S ribosomal RNA primers (*e.g.*, *Stat et al., 2017*) and continued development of reference databases may be necessary to properly assess regional diversity. However, no COI sequences are available for several of the most common isopod, ostracod and tanaids that occur at the site, even for congeneric species (*Austrosignum*, *Philomedes*, *Scleroconcha* and *Nototanais* sp.), and could explain the high proportion of reads not classified in the sediment. Similarly, annelids were the most diverse phylum (24 distinct taxa), including 11 of the 16 taxa with the highest mean relative abundances (Fig. 6), but only three were resolved to genus or species level. These taxa should be the focus of new DNA barcoding campaigns to improve their coverage in reference DNA sequence databases like BOLD. Until additional reference sequence data is available, these taxa can only be detected and identified to species or genus-level by morphology-based approaches, further emphasising the need for complementary approaches (*e.g.*, *Sinniger et al., 2016*).

The ideal eDNA metabarcoding marker is universal for the taxa of interest (*i.e.*, no major taxonomic primer bias) with sufficient sequence divergence to resolve species (*Clarke et al., 2014*), whilst being amenable to sequencing on widely used platforms such as the Illumina MiSeq. The three most commonly used metazoan metabarcoding markers target the nuclear 18S rRNA gene, or the mitochondrial COI or 16S rRNA genes. Reference sequence data should also be available for the marker of interest. Of the 64 benthic arthropod species recorded near the Davis and Casey East Antarctic stations, approximately half have sequence data available for COI or 18S at the genus level (31 and 29 species, respectively), but only a third have 16S data (22 species). Species-level data for these benthic arthropods is only 7–11 species per marker. However, the lack of reference sequences for any marker can be overcome with a dedicated sequencing campaign, or an OTU-based approach to taxonomy (*e.g.*, *Ji et al., 2013*). Although an OTU-based approach is possible in the absence of reference sequence data, species-level identifications are by far the most desirable for biodiversity monitoring. Researchers wanting to integrate DNA-based data with data obtained through other methods like morphology-based identification should target markers that provide species-level resolution, such as COI, 16S regions >100 bp in length, and/or universal 18S V4 markers (Table 1).

Metabarcoding markers should also be targeted to the taxa of interest, with the amplification of non-target reads kept to a minimum. We found only ~40% of reads were assigned to metazoa using the Leray COI marker, and a significant proportion of reads were lost to non-target taxa (both non-metazoan and unclassified). Low proportions of metazoan reads could be addressed by concentrating metazoan taxa within the water or sediment sample by filtration or elutriation, for example (*e.g.*, *Brannock & Halanych, 2015*; *Suter et al., 2020*).

Comprehensive benthic metazoan biodiversity surveys may require re-designing the Leray COI primers to better target benthic arthropods and molluscs, or testing primers targeting the variable regions of the nuclear 18S or mitochondrial 16S ribosomal RNA gene. Although designing and testing new primers is beyond the scope of the current study, further investigations could include: (1) *in silico* PCR to test taxonomic resolution in closely related species (*e.g.*, congeneric species), and to test for non-specific amplification
of other common environmental species or contaminants (bacteria, human DNA); (2) testing primers on DNA extracts from individual specimens to optimise PCR conditions and confirm a PCR product of the expected size is amplified with no non-specific amplification; (3) a small-scale sequencing run testing primers on a mix of DNA extracts from benthic arthropods and a small number of environmental samples.

## CONCLUSIONS

We recommend future DNA-based biodiversity assessments for Antarctic nearshore environments include water eDNA along with sediment sampling if suitable for the research or monitoring question, given similar numbers of species were detected in both water and sediment samples, the comparative ease of collecting water samples, and the versatility of sampling both hard- and soft-bottom sites. DNA in sediments may be derived from dead tissue, such as seaweed wrack, that originated from spatially or temporally distant sites, particularly given the temperature in Antarctic waters. Future eDNA studies testing the environmental drivers of eDNA community composition should collect data for environmental variables known to be important in structuring benthic communities (*e.g.*, sediment grain size, proportion of hard versus soft habitat, etc.) in conjunction with sample collection. We found that combining data from replicate water eDNA samples increased the number of taxa detected at each site. As well as being essential for rigorous statistical analysis, including replicates is probably a more efficient method for improving metazoan biodiversity detection than increasing sequencing depth for this marker, as the number of species detected per sample approached an asymptote with 1000 metazoan reads, and only ~40% of reads were assigned to metazoa.

Non-invasive, "capture-free" approaches like eDNA also raise the question: how do we know an organism detected with eDNA is really there? (*Darling, 2020*). Although several recent studies suggest that marine eDNA signals are a snapshot of the local community (*Jeunen et al., 2019*; *Jeunen et al., 2020*; *Suter et al., 2020*), the fate and transport of eDNA in Antarctic benthic environments poses unique challenges. Year-round low temperatures potentially extend the lifespan of eDNA. However, DNA degradation rate is not solely a physical property, as both the water column and sediment contain active microbial communities. Low current velocities in many coastal Antarctic sites reduce the chance that DNA will be introduced from distant sites, but could increase residence time beyond that of the parent organism. For example, the higher relative abundance of the sea-ice associated copepod *Paralabidocera grandispina/antarctica* in sediment compared to water eDNA is likely to reflect accumulation of eggs in the sediment (*Swadling et al., 2004*), demonstrating that DNA-based detections do not necessarily reflect species active within the sample. An understanding of the spatial distribution of eDNA in Antarctic benthic ecosystems, from a few metres to several kilometres, should be paired with careful cataloguing of the actual biodiversity present using other methods. Knowledge of spatial and/or temporal eDNA variation could then be combined with site occupancy-detection models to accurately infer species distributions (*Chen & Ficetola, 2019*).

## ACKNOWLEDGEMENTS

James Marthick (Menzies Institute for Medical Research, University of Tasmania) facilitated use of the MiSeq Genome Sequencer. Helen Achurch (AAD) helped adapt the map in Fig. 1. Ben Raymond (AAD) provided advice on statistical analysis.

### Funding

Field work and laboratory analyses were supported through Australian Antarctic Program Project 5097, lab work was partly funded through the Australian Antarctic Science Program (AAS-4556). The funders had no role in study design, data collection and analysis, decision to publish, or preparation of the manuscript.

### Grant Disclosures

The following grant information was disclosed by the authors:
Australian Antarctic Program Project: 5097.
Australian Antarctic Science Program: AAS-4556.

### Competing Interests

The authors declare there are no competing interests.

### Author Contributions

- Laurence J. Clarke conceived and designed the experiments, performed the experiments, analyzed the data, prepared figures and/or tables, authored or reviewed drafts of the paper, and approved the final draft.
- Leonie Suter analyzed the data, prepared figures and/or tables, authored or reviewed drafts of the paper, and approved the final draft.
- Bruce E. Deagle conceived and designed the experiments, authored or reviewed drafts of the paper, and approved the final draft.
- Andrea M. Polanowski, Glenn J. Johnstone and Jonathan S. Stark conceived and designed the experiments, performed the experiments, authored or reviewed drafts of the paper, and approved the final draft.
- Aleks Terauds analyzed the data, authored or reviewed drafts of the paper, and approved the final draft.

### Field Study Permissions

The following information was supplied relating to field study approvals (*i.e.*, approving body and any reference numbers):

Field work permits were granted by the Australian Antarctic Division (ATEP 19-20-5097).

### Data Availability

The raw sequence data are available at NCBI: PRJNA720767. The supplementary files include MID tags, zOTU table and a final species table.

## Supplemental Information

Supplemental information for this article can be found online at http://dx.doi.org/10.7717/peerj.12458#supplemental-information.

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
