# Peer review of "Environmental DNA metabarcoding for monitoring metazoan biodiversity in Antarctic nearshore ecosystems"

_PeerJ, doi:10.7717/peerj.12458_

## Round 0.1 · original submission · Major Revisions

Dear Laurence and co-authors,

I have received three, rather contrasting, independent reviews of your study. Apologies for the delay. While all reviewers clearly recognised the quality/novelty of your work, they have collectively raised a number of issues that will need to be addressed in your revised manuscript. One fundamental limitation that was highlighted by two reviewers is the use of only one marker (COI) for this study. While I don't think that the use of a single marker automatically warrants rejection, I do agree that the use of COI for characterizing Antarctic biodiversity is problematic in this case. Please consider adding another marker (18S rDNA gene?) on a subset of samples if you can - or - clearly acknowledge the limitations raised by the reviewers and rethink the focus of the paper. Other issues relate to the Methods section which needs to be improved.

Overall, the reviewers have provided you with excellent suggestions on how to improve the manuscript, and I'll be looking forward to receiving your revised manuscript along with a point-by-point response to the reviewers comments.

With warm regards,
Xavier

Reviewer 1 ·

Basic reporting

The manuscript is clearly written, topic is relevant and important to pursue.

Experimental design

Overall, I like how the study was conducted. However, just because the BOLD databases contain 200k sequences does NOT mean the COI marker is appropriate for Antarctic biodiversity investigations. Most of the sequences in BOLD are not Antarctic...nay, the LARGE majority. Many studies are showing now that marker choice is taxonomic group-dependent and studies recommend using multiple markers (COI, 16S, cytb, etc.) to get a better overview of total biodiversity, which is the goal of this study.

One other note is that I very much like the authors comments on field and laboratory blanks/negative controls.

Why use only half the filter? Studies have shown that the DNA is heterogenous....what if youre missing the targets/more interesting species on the other side of the filter?

Validity of the findings

Valid findings. See design for comments on improving/expanding the reach of these results.

One comment: the finding of 77-80% similarity to "cyclopoida sp." is extremely variable. That difference could be the difference between cerolids and sea spiders. This circles back to my "additional markers would be of use" comment from above.

Additional comments

Overall a nice manuscript that shows eDNA should be expanded in Antarctic waters.

Reviewer 2 ·

Basic reporting

Please see general comments. The main questions and main focus are the article are not clearly presented. They are several competing "main" points/aims.

Experimental design

Please see general comments. Experimental design is flawed. Using only one genetic marker is not up to current standards. There is also a lack of justification for use of the marker.

Validity of the findings

See General comments - Some of the experimental issues make the findings questioned

Additional comments

This manuscript uses the COI marker to explore eDNA diversity in water and sediment Vestfold Hills in the Eastern Antarctic. There are several positive aspects this study (e.g. looking at both sediment and and water column, considering hard substrate). However, there are many issues that are just not up to current standards and the how the central ideas and focus of the paper are presented can be confusing to the reader. While the topic is of broad interest, the execution of this work renders it problematic for publication. Reasons for this rather negative assessment are below.

Marker choice —One marker is not sufficient by current standards. This is a major issue and second (or ideally at least 2 more) should be used prior to publication. All markers and primers have their issues but COI is particularly problematic for metabarcoding (which is done here) versus eDNA detection analyses for particular species. Which COI primers are used is never stated (making the work non-repeatable). This is critical in that COI primers are notorious for taxonomic bias. Also many are degenerate requiring touchdown PCR as done here. Agreed that low annealing is a problem but touchdown PCR also causes issues. Taxonomic bias is likely the reason several abundant arthropod taxa are not detected. However, that possibly of primer bias is not discussed. Moreover very little justification is provided for use of COI. At the end, the manuscript the Leray & Knowlton 2016 paper is used to argue that COI is preferred because it has better resolving ability than 18S. This statement is simply not as correct and Leray & Knowlton suggest. They are confused on the issues. Most of the 18S rDNA regions used for metabarcoding span variable regions (V1-2, V4, V9). In contracts, 18S was used for years for higher level analyses, but in those higher levels analyses these variable regions where usually excluded because they were too variable. The bottom line is that paper has to provide information and justification on the primers used, and importantly more than one marker should be employed.
Read number and sequencing depth. Only 5.45 million reads total were used. Why this is the case is not clear as even a single Miseq run produces more than 44 million paired end reads. The manuscript uses ratification to standardized read number which is good, but the number of 1000 is very low and it is not clear why. The 5.45 million is not bad by the cutting the rarifaction curve off at 1000 (or more to the point the inflection point being at about 100 reads) is just not consistent with what others see in sediment samples (see papers from Creer lab, Halanych lab, Bik lab, etc.)
Focus of the paper is not clear. In the abstract the main aim is to develop methods, but in the introduction the aim of the paper is to develop methods, characterize Antarctic samples (line 74), and address 3 questions (line 96 - note that the first question is 3 questions by itself - so it is really 6 questions) and then the Discussion leads with main finding about the choice of sample type matters. To be direct, this manuscript does not develop new methods. Everything here has been done before. Also the results that sample type matters (sediment versus water) is not novel. The manuscript needs to find a coherent and novel story. Consider focusing on what taxa were found in Eastern Antarctica or comparing water and sediment (but see next point).
Quantitative statements based on metabarcoding is challenging. How can a sample size from a Niskin bottle be compared to an amount of sediment? Throughout the manuscript there are statements about more OTUs found in sediment. That is true but some caveats should be clearly stated next to such statements. Also how sediment is processed is important. The manuscript focuses on small amounts of sediment similar to what is done with bacterial metabarcoding but see Brannock and Halanych 2015 (Marine Genomics 23:67).

Other points include:

Line 39 - Clitellates are derived polychaete - that is the node on the tree of life that defines polychaete is the same as the one that defines annelids. Thus the 10% comment is likely not correct.

Line 191 - creating a phylogenetic tree using Geneious -This is not a proper methodological description. Also a tree across metazoans with 300bp of COI. Not sure why this adds anything.

There is no discussion of DNA from live taxa versus historic DNA (mainly a sediment issue).

For the water samples one of the primary environmental variables is likely to current regimes, especially given the narrow network of channels and fjords in the Vestfold Hills region.

There is no comment on the non-metazoan taxa which compose roughly half the sample. The fact that so many sequences are non-metazoan raised further questions about the primers.

Main discussion point does not track main questions. Second sentence a question

Line 341 Water or sediment is hardly a “niche;” it is a habitat.

·

Basic reporting

The manuscript is well written. A couple of recent relevant references could be cited. The raw data is shared in the SRA repository, but it is missing some crucial metadata (sample tags) which prevent the reader from reproducing the pipeline. This metadata could be added as supplementary material.

Experimental design

The research questions are well defined, But some methods could be described in higher detail. The bioinformatic procedures could be improved by performing a minimal abundance filtering step.

Validity of the findings

The findings are generally valid and the implications of the results are well discussed. There is no speculation.

Additional comments

This manuscript by Clarke et al. is an interesting contribution to our scarce knowledge of molecular biodiversity of Antarctic benthic ecosystems. However, I think that there are a few significant issues that need to be addressed before this manuscript is ready to be publishable in PeerJ.
- First, the materials and methods are not completely clear and some details should be clarified (see below).
- Second, the data availability can be improved. Although the raw reads datasets (2x250 bp) have been uploaded to the NCBI SRA repository, the information in this BioProject is not enough to reproduce the bioinformatic procedures. The authors have used 6-bp sample tags, but the information of these tags can not be found anywhere. I suggest that the authors should upload a table of the tags used for every sample as supplementary material to this paper. Otherwise, all the data shared in the SRA repository would be mosly useless. Moreover, the final table of zOTUs (after denoising) and the final curated table of species (after taxonomic clustering) should also be uploaded and shared as supplementary material to this paper.
- Third, I am a little concerned that the authors did not mention to use any filtering step based on minimal read abundance by sample in their bioinformatic pipeline. But then they are performing calculations (beta diversity) based on presence/absence. For this calculations to be robust, it is crucial to establish a minimal read abundance filter to consider a true presence. Otherwise, false positives are very easily spread along the dataset (for example, due to tag jumping), which can significantly interfere with presence/absence metrics. I would be more comfortable if a minimal read abundance filtering had been used before the ecological analyses. This minimal abundance filtering should be ideally done in a sample-by-sample basis, in order to remove low-frequency MOTUs in every sample, before calculating the rarefied species table.

Other suggestions and issues that should be addressed:

INTRODUCTION
L81: While water eDNA can -potentially- be used to study both soft- and hard-bottom communities, two recent studies (which could be cited here) have highlighted the limitations of this approach. (1) Antich et al. (2020, doi:10.1111/mec.15641) proved that less than 5% of hard-bottom benthic organisms from Mediterranean shallow communities could be detected from water eDNA, even when water was taken very closely to the bottom. (2) Brandt et al. (2021 doi:10.1038/s41598-021-86396-8) showed that eDNA from deep-sea water did not either reflect the composition of the benthos (3-8% benthic MOTUs detected in water), even when high volumes of water were filtered. Both papers conclude that sampling water is not a useful alternative for characterising benthic diversity.

MATERIALS AND METHODS
L141: Change "(approximately 300 bp)" to "(313 bp for most metazoan species)".

L144. The sequences of the metabarcoding primers used for the amplifications should be written here explicitly. My guess is that the original Leray et al. (2013) primers were used, with inosines in the reverse jgHCO2198 primer and with no inosines in the forward mlCOIintF primer. But it is important to state this detail clearly, for future reproducibility.

L166: Are the authors sure that they used a V2 Nano kit? This kit yields barely a maximum of 1 million raw reads. It is impossible to get 8 millions of paired-end raw reads (as deposited in BioProject PRJNA720767) or 5.45 million paired-end reads (after filtering and QC) using a Nano kit. I think that they used either a V2 full kit (maximum yield ~15 million raw reads) or several Nano kits (less probably, since this is not economically optimal). Please double-check this.

L180-205: The authors are using a hybrid system to get their diversity matrices, which is not totally clear to me, and should be further clarified here. First, they generate the zOTU table with UNOISE3, and this is clear. Then, I understand that taxonomic clustering is used to delimite species and generate a table at the species level. This is a reasonable way to go, as most ecological analyses with COI should be done at the species (cluster) level, rather than at the zOTU level (Antich et al. BMC Bioinformatics, 2021). But the procedure to perform the taxonomic clustering should be explained more clearly. Authors mention that they are using BLASTN against the NCBI, then they use MEGAN to assign taxonomy. Why using BLASTN then? Was BLASTN used to filter non-metazoans sequences, and then MEGAN used only to assign the remaining metazoan sequences? This should be clarified. Finally they generate "a phylogenetic tree" with Geneious for further clustering other sequences at >97% identity. They should explicitly mention which phylogenetic algorithm for tree generation was used, since Geneious has many different possibilities. Moreover, It would be a good idea to include in the supplementary materials the raw table at the zOTU level and the final curated table at the species (taxonomic cluster) level. Also, a summary of the total number of zOTUs in the initial table and the total number of species-level clusters in the final table should be more clearly mentioned in the Results (currently there are two sentences that are contradictory). This would help to better understand the results of the non-orthodox taxonomy clustering procedure used.

L214: Change "nut" to "but".

L208-209: It is not clear to me is this "species richness" rarefaction and the following analyses was tested at the zOTU level or at the species cluster level. Which of the two biodiversity tables was used for these analyses? Only the 99 assigned Metazoan table? Or the whole zOTU table? Please specify.

L219-222: The same is true for the beta-diversity analyses. The results of both Jaccard and Bray-Curtis dissimilarities can be very different if they are performed at the zOTU level or at the "species" level. The ecological analyses for COI should be done using the species-level matrix, and not the zOTU matrix. I guess that the authors used this. But, please, specify.

RESULTS
L245: Between the 5.45 million raw reads and the 3950 zOTUs, it is also interesting to know how many unique sequences were there before denoising. I think this number should be mentioned here.

L245-246. "518 zOTUs" containing 2.16 million reads are assigned here to 99 metazoan taxa. However in L254-255, it is said that the 99 metazoan taxa contained "479 zOTUs" (and 2.16 million reads). These two sentences are contradictory and contribute to generate confusion. Please double-check the numbers.

L279 and L309-310: In my experience, using sqrt-transformed (or even fourth-root transformed) read abundance data can significantly improve the ordination of Bray-Curtis dissimilarities for COI metabarcoding data. Have you tried if you could get more logical ordinations of the replicates after any of these transformations? Just out of curiosity...

DISCUSSION:
L324: You can also cite Antich et al (2020) and Brandt et al. (2021) here. Both papers are very relevant for this issue.

FIGURES:
FIG5: The caption of Fig. 5 is incomplete. (A) is described but (B) is not.

Congratulations to the authors for a nicely written, highly interesting, paper.
Owen S. Wangensteen

---

## Round 0.2 · Major Revisions

Dear Dr. Clarke and co-authors,

I have received three reviews of your revised manuscript. Unfortunately, two reviewers now voted to 'Reject'. In a situation like this, I would normally have to follow the majority rule and reject the manuscript. However, after careful examination of the revised manuscript and rebuttal, it appears that most methodological issues raised by the reviewers from the original version have been adequately addressed. The conflict at play here comes from the use of a single marker (COI) and the perception of two reviewers that you made a limited attempt to truly embrace some of the original comments and address the issues raised (i.e. more fully acknowledge the limitations of using a single COI marker and expend the Discussion with a more balanced view on this issue).

My view remains that this is an interesting and good quality study that explores Metazoan diversity from understudied environments in Antarctica which would constitute a valuable contribution to the scientific community - the current content meets the PeerJ editorial criteria and the journal does not stipulate anywhere that the use of a single marker automatically warrants rejection based on that alone. At the same time, one need to recognize that there is no 'silver bullet' marker and COI is no exception. If the message of the study is to be as accurate as possible, then there is no way a single marker would work. But if the goal is to explore a new environment and compare biodiversity estimates between sample types (in your case differences between sediment and water samples or among water depth), then I have no objection of using COI for that purpose.

Where to from here? Looks like we are at an impasse, but it is my hope that we can find a collegial solution. I would like to give you another opportunity to respond to Reviewer#1 and #2. What I need from you is to:

1. consider addressing some of the original comments in a more adequate fashion. In particular by stressing the preliminary/exploratory nature of this study and expending the Discussion on the limitations of using a single marker. This could be done relatively easily by adding a dedicated section on the pros and cons of commonly used metabarcoding markers (COI, 18S, 16S, etc..). However, I cannot guarantee that this will satisfy the reviewers, although it is my hope that, with some care, it will.

and/or

2. consider obtaining some additional (even limited) data from another marker as a point of comparison. This option will certainly satisfy the reviewers and I would be happy to grant you the necessary time extension may you decide to go down this route.

However, I do recognize that this manuscript has been under review for a considerable amount of time (>4 months), and therefore I would understand if you decided to submit the study to another journal. If so, please do let me know of your decision.

With warm regards,
Xavier

Reviewer 1 ·

Basic reporting

The authors note in their text that COI is the 'best' marker for use in this work. However, as reviewed previously, COI fails to detect many species and multiple marker studies are commonplace in the eDNA literature. Simply stating the authors believe COI is the best without testing is insufficient.

Experimental design

Single marker eDNA studies, particularly using COI as their only marker, is not sufficient to make any biodiversity conclusions. See for example: https://www.mdpi.com/2073-4441/13/13/1767. This is one of many papers out that discuss this. While in a different system, COI alone will not work as was noted in previous reviews. Other markers could be either added to the databases or simply used and considered as operational taxonomic units (OTUs).

Validity of the findings

With only a single marker, as previously reviewed, these findings are not meaningful, are speculative, and cannot be considered robust or sound.

Reviewer 2 ·

Basic reporting

The original manuscript received three reviews with some agreement on the strengths and weaknesses of the work. There is considerable excitement for the topic. Some of the comments made focused on design flaws in the analyses (e.g., one marker, read depth, rarefaction). The revision was far more superficial than these comments deserved. There are limited wording changes to the manuscript to try and explain a particular view rather than address the underlying problem or presenting a fair presentation of the weaknesses of the approaches used.

Experimental design

see below

Validity of the findings

see below

Additional comments

The use of the single COI marker is still problematic and rather than be direct about some of the short comings, the manuscript tries to defend this weakness. For example, CO1 is defended because it is has been the most sequenced in terms of the number of taxa…agreed. However, that does not make it good enough to use by itself. The explanation of the 18S variable regions is more complex than just saying Leray and Knowleton say, (more or less) “18S variable regions do not work”. Variable regions due underestimate diversity, but so does COI. That is one reason people use more than one marker. Also, the degree of underestimation is taxon specific for both markers, and for some taxa 18S variable regions work just fine. The explanation presented does not justify why COI alone is enough. As another example, the Yang et al paper is mentioned as showing that wet lab and bioinformatics can be used to limit OTU drop out. However, that paper employed a pool of only arthropod sequences with different primers. The limited drop out by Yang et al. is not comparable given the methods used here. Also, the over reliance on the Leray et al primers has not been adequately discussed. As a comparison point, Folmer et al 1994 primer were claimed to work over across animals, but that turned out not to be true. There are about three or four other sets of COI primer papers that have made similar suggestions, none were correct. The manuscript should embrace and discuss the issues and challenges with this marker instead of trying to explain it away or ignore it altogether.

To be clear just because “we feel that COI is by far the best target” does not justify its use as the only marker.

The Leray et al and the Leray and Knowlton papers are generally very good, but they over interpret some issues which are be amplified here.


Most of the other suggestions were adequately addressed.

·

Basic reporting

No comment.

Experimental design

No comment.

Validity of the findings

No comment.

Additional comments

The manuscript has been significantly improved with the additional changes made by the authors. I am now fully satisfied with this version, and I strongly recommend it to be published in PeerJ.
I would like to add that I strongly disagree with the views expressed by Reviewer 2, who is asking for the use of at least two markers, and questioning the validity of these findings only because they are exclusively based on COI. In my opinion, we can indeed extract highly valuable taxonomical and ecological information from metabarcoding data sets based exclusively on COI. We should never reject a paper based only on this fact. I fully agree with the authors' rebuttal. In my experience, in most cases, 18S metabarcoding provides only low-resolution data for metazoans (typically at the family or order level), which does not add much value to the information that you can retrieve from COI. 18S is probably more important when targeting whole Eukaryotic diversity patterns at coarse taxonomic resolution (e.g. groups of protists), but it is definitely not needed for metazoans at the species level. Moreover, samples from Antarctica are always precious, and any new information that we may retrieve from them, even from a single COI marker, is definitely worth publishing!
Again, congratulations to the authors.

---

## Round 0.3 · accepted · Accept

Dear Dr Clarke and co-authors,

I am pleased to accept your revised manuscript for publication in PeerJ. I very much appreciate your efforts in fully addressing the reviewers' comments, in particular, the addition of the in-silico primer analysis and modified discussion which have greatly improved the manuscript.

Thank you for your nice contribution to the eDNA field.

With warm regards,
Xavier

·

Basic reporting

No comment

Experimental design

No comment

Validity of the findings

No comment

Additional comments

I am completely satisfied with this version of the manuscript. I think that the additional ecoPCR analyses performed by the authors are justifying the choice of metabarcoding primer. This additional information may be useful for other researchers to design metabarcoding projects in the future.
I am happy with the editor's decision of not rejecting this work based only on the false premise that using a single marker completely invalidates any retrieved information. The presented results are indeed novel, interesting, and unique. And they contribute to increase our scarce knowledge on molecular biodiversity of Antarctic marine ecosystems.
I strongly recommend the publication of this work in PeerJ.